# I Told You So! Verbal cue beliefs are associated with truth detection, but not lie detection

Glynis Bogaard[ORCID][1]*, Sarah Elizabeth Fieweger[2], Ewout H. Meijer[1]

**1** Department of Clinical Psychological Science, Faculty of Psychology and Neuroscience, Maastricht University, Maastricht, The Netherlands, **2** School of Psychology and Humanities, University of Lancashire, Preston, United Kingdom

* glynis.bogaard@maastrichtuniversity.nl

## Abstract

This study aimed to 1) replicate and extend previous findings on the relationship between accurate beliefs about the diagnosticity of verbal deception cues and actual truth/lie discrimination as reported in Bogaard and Meijer (2017), and 2) examine the role of presentation modality. Participants (N = 246) listed their beliefs about deception cues and then judged the credibility of truthful and deceptive autobiographical statements presented in one of three modalities: audio-only, transcript, or audio-visual. As in the original study, participants provided continuous credibility ratings. In addition, we asked participants to make binary truth/lie judgments, allowing us to assess discrimination accuracy. We replicated the original finding, namely a small but significant positive correlation between accurate verbal cue beliefs and the credibility ratings of truthful, but not deceptive, statements. A small association was also observed between accurate cue beliefs and truth discrimination accuracy. Contrary to our expectations, modality did not significantly influence truth/lie discrimination or credibility judgments. These findings suggest that while cue knowledge may support the recognition of truthful statements, it does not aid in the identification of lies, nor does the presentation format substantially impact veracity judgments.

## Introduction

People across the world tend to believe that liars avoid eye contact, shift their posture, touch their face or body, and appear nervous [1]. These beliefs center on observable nonverbal behaviors and are shared not only across cultures but also professional domains (i.e., police officers, teachers, and social workers) [2]. However, a substantial body of research has shown that these commonly endorsed cues to deception are not valid indicators of lying. In a comprehensive meta-analysis, DePaulo, Lindsay [3] examined 158 cues across 120 studies and found that nearly 75% of them were unrelated to deception. Of the few cues that did show a relationship, most were weak. These findings challenge the intuitive appeal of nonverbal

**Data availability statement:** The data underlying this study are deposited in Dataverse and are openly accessible at https://doi.org/10.34894/CY17PX.

**Funding:** The author(s) received no specific funding for this work.

**Competing interests:** The authors have declared that no competing interests exist.

behaviors as markers of deceit, and may explain the well-documented finding that human truth/lie discrimination accuracy hovers only just above chance level, at approximately 54% [4].

In contrast to nonverbal cues, verbal cues appear to be more promising indicators. Verbal cues refer to specific characteristics embedded in the content of what people say, such as the amount of detail, clarity, coherence, and internal consistency. Research has shown that liars tend to provide fewer details and are more prone to contradictions than truth tellers (DePaulo et al., 2003). Such verbal cues are more diagnostically valid than nonverbal behaviors, with moderate effect sizes [5,6]. Furthermore, lie detection tools based on verbal content have demonstrated higher classification accuracy, sometimes reaching average success rates of around 70% [7]. In addition, individuals who are more proficient at detecting lies tend to rely more on verbal content than on nonverbal behaviors [8].

One proposed way to improve deception detection is therefore to reduce the availability of nondiagnostic nonverbal cues and instead encourage focus on content. Removing or limiting visual information may help shift attention toward more diagnostic verbal indicators. Hence, the format in which a statement is presented – whether through video, audio, or written transcript – can therefore play an important role in deception detection. Previous research comparing lie detection performance across audio-visual, audio-only, and transcript-based conditions found that performance was lowest in the audio-visual modality, suggesting that the presence of visual cues may indeed distract from more reliable verbal and vocal features [9]. Similarly, research found that access to visual information increased the likelihood of both false positives and false negatives, likely due to cognitive biases triggered by ambiguous behaviors [10]. In line with these findings, more recent research reported that audio-only presentations led to higher accuracy rates than audio-visual presentations, particularly for real transgressions [11]. However, audio-visual presentations do seem to outperform visual-only modalities for spontaneous lies [12]. In contrast, Evanoff, Porter [13] compared video to transcript formats of high-stakes emotional lies and found that overall truth/lie discrimination accuracy across modalities did not differ significantly. Bogaard and Meijer [14] similarly compared audio-visual and transcript conditions for negative autobiographical narratives and reported no significant difference in overall truth/lie discrimination accuracy.

Taken together, these studies offer inconsistent evidence about the impact of modality on truth/lie discrimination performance. This variability suggests that factors beyond presentation format may be critical in explaining successful truth/lie discrimination. One such factor is individuals' understanding of which cues are diagnostic. Even in the absence of distracting visual information, people may still struggle to detect deception if they rely on incorrect or stereotypical cues. This raises the question of whether knowledge about valid verbal cues can improve truth/lie discrimination. Earlier studies investigated this by examining a combination of verbal and nonverbal cues and reported positive associations between cue knowledge and veracity judgment accuracy for economics students [15], psychology students [16], students interested in social psychological studies and personnel professionals [17].

Bogaard and Meijer [14] specifically examined whether people's beliefs about the diagnosticity of verbal cues such as detail richness, number of contradictions, and the coherence of a statement relate to how they judge the credibility of truthful and deceptive statements. In their study, undergraduates and police officers rated the credibility of video and transcript-based autobiographical statements using a 10-point scale (1 = completely deceptive; 10 = completely truthful). Participants also indicated which verbal cues they believed to be associated with deception. The researchers then examined the correlation between the number of correctly endorsed verbal cues and credibility ratings, separately for truthful and deceptive statements. Their findings showed a positive correlation between accurate cue beliefs and credibility ratings of truthful statements, but no such relationship for deceptive ones. In other words, better knowledge of valid verbal cues was associated with better recognition of truths, but not better detection of lies.

The current study aimed to replicate and extend the findings of Bogaard and Meijer [14] in two ways. First, we examined whether participants with more accurate beliefs about verbal cues would be better at judging the veracity of statements. In the original study, only continuous credibility ratings were collected, but in the current study, we additionally asked participants to make a binary truth vs. lie judgment for each statement. This allowed us to assess not only the relationship between cue beliefs and credibility judgments, but also their association with truth/lie discrimination accuracy. Based on previous research, we hypothesized that higher endorsement of valid verbal cues would be positively associated with both outcomes, especially for truthful statements (Hypothesis 1). Second, we investigated whether the modality in which statements were presented (audio-only, transcript, or audio-visual) influenced participants' truth/lie discrimination accuracy. Drawing on earlier findings that suggest visual information may impair truth/lie discrimination accuracy, we hypothesized that participants in the audio-visual condition would perform worse than those in the audio-only or transcript conditions (Hypothesis 2).

## Materials and methods

### Preregistration and data availability

The study design, hypotheses, and planned analyses were preregistered prior to data collection on the Open Science Framework, see here https://osf.io/xsf9m/overview. In our preregistration, we used the term *lie detection accuracy* to refer broadly to participants' ability to distinguish between truthful and deceptive statements. However, this terminology does not fully capture the specific outcomes assessed in our study, nor the design of the original study we aimed to replicate [14]. Therefore, we have adopted the term *truth/lie discrimination accuracy* to better reflect the focus on participants' ability to distinguish between truths and lies overall, as well as their performance on truthful and deceptive statements separately. Note that this change in terminology only reflects a refinement in how we describe the outcomes, not a deviation from our original analytical plan.

The data underlying this study are deposited in Dataverse and are openly accessible at https://doi.org/10.34894/CY17PX.

### Design

The current study used a 3 (Modality: audio-only, transcript, audio-visual) x 2 (Veracity: truth vs lie) mixed design with modality as a between subjects factor and veracity as a within subjects factor.

### Participants

Based on an a priori G*Power analysis for a 3 × 2 mixed-design ANOVA (within-between interaction), a minimum of 111 participants was required to detect a small-to-moderate effect size (f = .15) with a power of.80, using estimates from Bogaard and Meijer [14]. In addition to this participant-level requirement, we also considered Levine's [18] recommendation that at least 500 judgments per condition are necessary to make reliable inferences about truth/lie discrimination accuracy. Given our 3 (modality: audio-only, transcript, audio-visual) × 2 (veracity: truth vs. lie) design, and that each

 

participant made eight veracity judgments, a minimum of 188 participants was needed to achieve approximately 500 observations per modality condition.

To meet both the participant-level and modality-level thresholds, we recruited a total of 246 participants. Recruitment took place internationally through social media platforms, our university's research participation system, and Prolific. Undergraduate students received course credits, while non-students were compensated with a monetary voucher or through Prolific. Of the 246 participants, 49.6% identified as male, 48.4% as female, and 2% as "other" or chose not to disclose their gender. Their age ranged between 18 and 64 years old, with a mean age of 26.59 years ($SD = 7.74$).

This study was approved by the Ethics Review Committee of the Faculty of Psychology and Neuroscience (Maastricht University), under approval code ERCPN-233_13_02_2021. All procedures were conducted in accordance with the Declaration of Helsinki. All participants received a written information letter describing the aims of the study, procedures, potential risks, and benefits. After reviewing this information, participants provided written informed consent prior to participation. Data was collected between 11 February 2023 and 1 June 2023.

## Statements

To assess truth/lie discrimination performance, participants were presented with eight video-recorded statements. The stimuli were developed using the same procedure as Bogaard and Meijer [14]. In each recording, an interviewee was shown seated, with their upper body (from the knees up) visible. The interviewer's voice was audible, but they were not visible on screen. All interviewees were acquaintances of the research assistants, which allowed for the verification of the truthfulness of each statement. During the interview, the following open-ended questions were asked:

"Can you tell me about a negative event you experienced?"

"How did you feel when this happened to you?"

"What did you do when this happened to you?"

Initially, seven individuals were interviewed about negative autobiographical events (e.g., an accident, a crime, a loss, etc.). Each person provided two accounts: one truthful and one deceptive. The order of the two types of statements was counterbalanced across participants. For the deceptive accounts, interviewees were instructed to describe a negative event as if it had happened to them, drawing from a plausible but fictitious scenario. Interviewees were informed that their recordings would be used for a study in which others would assess the veracity of their statements. This process yielded 14 statements: seven truthful and seven deceptive.

To ensure stimulus quality, the recordings were pretested with 11 undergraduate students. They rated each video on believability (1 = highly unbelievable, 5 = highly believable), audio quality, and visual quality (both on 5-point Likert scales, from 1 = very poor to 5 = very good). One interviewee was excluded due to low audio quality. No other interviewees stood out in any way and all six were retained.

To expand the stimulus pool, eight additional individuals were interviewed using the same procedure. These new recordings were again rated by 11 undergraduates using the same scales. All were deemed acceptable and included in the final stimulus set.

In total, the final set included statements from 14 senders, each providing both a truthful and a deceptive account, resulting in 28 statements. The average statement was 251.57 words long, and the mean duration of the videos was 95.39 seconds.

## Questionnaire

Cues were selected and categorized based on Bogaard and Meijer [14]. These cues were derived from two established content-based credibility assessment tools: Criteria-Based Content Analysis (CBCA; [19]) and Reality Monitoring (RM;

[20]). Specifically, we included all eight RM cues, and a subset of 12 (out of 19) CBCA criteria. Consistent with prior work [14], seven CBCA items were excluded because they are rarely observed in adult statements or are primarily relevant in child witness contexts [21]. Additionally, three overlapping items across the two tools were merged to avoid redundancy, resulting in 17 distinct verbal cues. For each cue, participants were provided with a brief definition and example and asked to indicate their belief about its relation to deception by selecting one of four options: "less used by liars", "no difference between liars and truth-tellers", "more used by liars", or "I don't know". See S1 Appendix for a full list of criteria and their explanation.

## Procedure

Once participants provided consent, they started the survey which was presented online. Subsequently, participants were presented with the deception cues and their description. After reporting their beliefs for each of the cues, the participants were randomly assigned to one of the three modalities (audio-visual, transcript, or audio-only) and shown eight statements (i.e., four true and four deceptive) from unique senders that were randomly drawn by Qualtrics from the pool of 28 statements. We used JavaScript to ensure that the same interviewee would not be presented more than once.

After each statement, participants were asked to rate the statements' credibility (1 = not credible; 10 = very credible). Next, participants were asked: "If you have to choose, do you find the statement truthful or deceptive?". Participants could answer this question by selecting "truthful" or "deceptive". Once all eight statements had been presented, the participants were given a list of the cues and their definitions again and asked to indicate which of the cues they had used to formulate their judgments by selecting "yes" or "no" for each cue. Lastly, participants were asked to respond to the statement "I was motivated to participate in this study" using a 5-point Likert scale (1 = strongly disagree, 5 = strongly agree) and "I was serious about my participation" with "yes" and "no" as possible answers. Participants were thanked and debriefed.

## Results

### Participant engagement

Participants completed two brief engagement checks. Most participants reported high motivation (57.7% selected "5" and 32.5% selected "4"), with an overall mean of 4.39 (SD = 0.93). Second, all participants (100%) indicated that they took the study seriously. These responses suggest the sample was engaged, supporting confidence in the quality of the self-reported data.

### Endorsement of cues and relationship with extant literature

Table 1 shows to what extent participants endorsed the directions of the presented cues. Data analysis is identical to Bogaard and Meijer [14]. First, we recoded directional answers as −1 indicating 'less for liars' and 1 indicating 'more for liars'. The neutral option was recoded as 0. The responses are presented as the percentage of participants who made each selection. The "I don't know" responses were omitted from further analyses. Next, we analyzed the data with one sample Wilcoxon signed-rank test for each cue to examine significant deviations from 0, which indicates a preference for a directional answer. Given the large number of tests, we adjusted the alpha level to .001. Results again showed that participants associated a high number of cues with deception (i.e., 10 out of 17 verbal cues).

For the cues, we used the same validity classifications as Bogaard and Meijer [14] and Bogaard, Meijer [22]. That is, a cue was considered diagnostic if prior empirical research showed it to differ significantly between truthful and deceptive statements in the expected direction in more than 65% of studies or if it met a minimum effect size threshold (e.g., $d \geq 0.50$). For RM items, diagnosticity was based on findings from DePaulo et al. (2003) and Masip et al. (2005). For CBCA items, we drew from the meta-analyses of Vrij (2005, 2008), Amado et al. (2015), and DePaulo et al. (2003). Based on these criteria, 10 cues were classified as diagnostic, and seven as non-diagnostic as the evidence for their association with deception was inconsistent.

**Table 1. Percentages of chosen alternatives for each presented cue and average mean value ranging between −1 (less for liars) and 1 (more for liars).**

| Cues | −1 | 0 | 1 | ? | M | SD | t | p | d | Actual | Perceived |
|---|---|---|---|---|---|---|---|---|---|---|---|
| Coherence* | 50.8 | 22.0 | 24.8 | 2.4 | −.27 | .84 | −4.91 | <.001 | −0.32 | < | < |
| Clarity* | 50.8 | 24.0 | 24.0 | 1.2 | −.27 | .83 | −5.11 | <.001 | −0.33 | – | < |
| Spontaneous corrections* | 15.0 | 24.0 | 59.8 | 1.2 | .45 | .75 | 9.48 | <.001 | 0.61 | < | > |
| Inconsistencies* | 6.1 | 14.6 | 76.0 | 3.3 | .72 | .57 | 19.47 | <.001 | 1.26 | > | > |
| Perceptual information | 32.5 | 30.5 | 30.9 | 6.1 | −.02 | .82 | −0.32 | .750 | −0.02 | < | – |
| Emotions | 29.7 | 36.6 | 26.8 | 6.9 | −.03 | .78 | −0.59 | .554 | −0.04 | – | – |
| Amount of detail | 35.0 | 26.8 | 36.6 | 1.6 | .02 | .85 | 0.30 | .764 | 0.02 | < | – |
| Spatial information | 31.7 | 34.1 | 30.9 | 3.3 | −.01 | .81 | −0.16 | .872 | −0.01 | < | – |
| Unstructured production* | 17.1 | 25.6 | 51.2 | 6.1 | .36 | .77 | 7.15 | <.001 | 0.47 | < | > |
| Description of interaction | 32.9 | 31.7 | 26.8 | 8.5 | −.07 | .81 | −1.24 | .217 | −0.08 | – | – |
| Temporal details | 32.5 | 35.4 | 28.9 | 3.3 | −.04 | .80 | −0.73 | .465 | −0.05 | < | – |
| Superfluous details* | 19.5 | 17.9 | 56.9 | 5.7 | .40 | .81 | 7.46 | <.001 | 0.49 | – | > |
| Reproduction of conversation | 32.1 | 31.3 | 27.2 | 9.3 | −.05 | .81 | −0.99 | .322 | −0.07 | < | – |
| Reconstructability | 32.1 | 27.2 | 33.3 | 7.3 | .01 | .84 | 0.24 | .814 | 0.02 | – | – |
| Unusual details | 30.9 | 20.3 | 43.5 | 5.3 | .13 | .88 | 2.31 | .022 | 0.15 | – | – |
| Plausibility | 38.2 | 27.6 | 28.9 | 5.3 | −.10 | .84 | −1.80 | .073 | −0.12 | < | – |
| Cognitive operations | 37.4 | 26.8 | 29.7 | 6.1 | −.08 | .84 | −1.48 | .139 | −0.10 | – | – |

*Note.* −1 = less used by liars than truth tellers, 0 = no difference between liars and truth tellers, 1 = more used by liars than truth tellers, ? = I don't know/no response. The correct beliefs regarding verbal cues are shown in the last column: > = more used by liars, - = no difference between liars and truth-tellers, < = less used by liars. Cues indicated with * significantly differ from 0 (*p* < .001). Bold indicates judgments in line with empirical evidence.

As can be seen in Table 1, for only 6 out of 17 verbal cues participants judged liars and truth tellers to differ. Of these six cues, only two were actually diagnostic of deception; coherence and inconsistencies. Moreover, of the 10 diagnostic cues, participants correctly recognized only two. Of the seven nondiagnostic cues, five were correctly identified as nondiagnostic, while two were erroneously associated with deception. Furthermore, the response distributions revealed considerable variability in beliefs across individuals, suggesting that there is little shared understanding of which verbal cues are truly diagnostic.

### Relationship between correct beliefs and truth/lie discrimination accuracy

Next, as studied in Bogaard and Meijer [14], we examined whether participants with more correct beliefs would exhibit better truth/lie discrimination accuracy (H1). Initially, each participant's number of correct beliefs regarding verbal cues was calculated. The average number of correct beliefs was 5.48 out of 17 verbal cues (*SD* = 2.35, range 0–12), which is an average 32.23% correct. Subsequently, the number of correct beliefs was correlated with the average accuracy, based on correct veracity scores, in addition to the average credibility of the truthful and fabricated statements.

The findings revealed a small but statistically significant correlation between holding accurate beliefs consistent with the literature and the credibility judgments of true statements (*r* = .136, *p* = .033). However, no significant correlation was found for credibility judgments of fabricated statements (*r* = .009, *p* = .883). This pattern was not reflected in the binary judgements: The correlation between accurate beliefs about verbal cues and accuracy for true statements was not significant (*r* = .119, *p* = .062), neither was the association for fabricated statements (*r* = −.029, *p* = .651).

### Accuracy and credibility judgments

Lastly, to investigate whether participants in the audio-visual modality have the lowest lie detection accuracy (H2), we first calculated their truth/lie discrimination accuracy from the forced choice question. Correct judgments were recoded as 1

and incorrect judgments as 0. Next, we summed up the correct lie judgments (max 4), correct truth judgments (max 4) and the correct total judgments (max 8). Then, we transformed this score into a percentage correct (e.g., for total accuracy this would be [Correct total judgments/ 8] *100). We also analyzed the Likert based credibility score; we averaged these credibility scores for the four deceptive statements and did the same for the four truthful statements. To examine whether modality influenced the accuracy and credibility of the judged statements, we conducted two 3 (Modality: audio-only, transcript, audio-visual) x 2 (Veracity: truth vs. lie) mixed ANOVAs with Veracity as the within subject factor. Once with lie detection accuracy as the dependent variable, and once with credibility judgment as the dependent variable.

For accuracy, the results revealed a significant main effect of Veracity, $F(1, 243) = 60.71$, $p < .001$, partial $\eta^2 = .20$, indicating that participants were significantly more accurate in judging truths than lies. There was no significant main effect of Modality, $F(2, 243) = 0.45$, $p = .637$, partial $\eta^2 = .004$, and no significant interaction effect between Veracity and Modality, $F(2, 243) = 0.08$, $p = .924$, partial $\eta^2 = .001$. See Table 2 for Ms and SDs.

For credibility, the results revealed no significant interaction effect between Veracity and Modality on truth/lie discrimination credibility, $F(2, 243) = 0.22$, $p = .801$, partial $\eta^2 = .002$. Additionally, the analysis revealed no significant main effect of Veracity, $F(1, 243) = 0.88$, $p = .349$, partial $\eta^2 = .004$ and no significant main effect of Modality, $F(2, 243) = 0.02$, $p = .977$, partial $\eta^2 < .001$. See Table 2 for Ms and SDs.

These findings provide no support for H2, as participants were not significantly worse at detecting lies in the audio-visual condition compared to audio or transcript.

## Discussion

The present study aimed to replicate and extend the findings of Bogaard and Meijer [14] by investigating the relationship between accurate beliefs about verbal cues and truth/lie discrimination accuracy, as well as the influence of modality on truth/lie discrimination accuracy. Two primary hypotheses were tested. First, we expected that participants who held more accurate beliefs about verbal cues would be better at truth/lie discrimination (H1). Second, we hypothesized that participants in the audio-visual condition would perform worse than those in the audio-only or transcript conditions (H2).

Regarding cue beliefs, similar to Bogaard and Meijer [14], we found that participants' beliefs were marked by considerable heterogeneity, reflecting a lack of shared understanding of which verbal cues are truly diagnostic. However, in the original study, participants correctly identified 60% of diagnostic cues (6 out of 10) and 43% of nondiagnostic cues (3 out of 7). In contrast, the current sample showed a more skewed pattern, correctly identifying only 20% of diagnostic cues (2 out of 10), but 71% of nondiagnostic cues (5 out of 7). Thus, although both studies show that laypeople's cue beliefs are selective and inconsistent, the current findings suggest an even stronger tendency to misidentify valid indicators of deception while being relatively accurate about cues that are actually nondiagnostic. These findings echo prior studies [2,14,22,23] demonstrating that laypeople hold diverse and largely inaccurate beliefs about indicators of deception.

This limited and inconsistent cue knowledge can explain the modest relationship observed between cue beliefs and truth/lie discrimination performance. In testing Hypothesis 1, we found weak support for the idea that accurate cue beliefs

**Table 2. Means and SDs of the accuracy (forced choice) and credibility (Likert Scale) scores for truthful and deceptive statements, separated per modality.**

| Condition | Truths | | | | Lies | | | |
|---|---|---|---|---|---|---|---|---|
| | Accuracy | | Credibility | | Accuracy | | Credibility | |
| | M | SD | M | SD | M | SD | M | SD |
| Audio-only | 60.71 | 26.34 | 5.93 | 1.39 | 39.29 | 27.11 | 6.03 | 1.45 |
| Transcript | 58.80 | 27.60 | 6.02 | 1.68 | 39.49 | 27.52 | 6.03 | 1.48 |
| Audio-visual | 58.24 | 28.03 | 5.91 | 1.58 | 36.64 | 25.69 | 6.09 | 1.50 |
| Total | 59.15 | 27.34 | 5.95 | 1.56 | 38.41 | 26.68 | 6.05 | 1.47 |

are associated with better lie detection performance. More precisely, we found a small but statistically significant positive correlation between accurate cue beliefs and the credibility ratings of truthful statements and the correlation with accuracy for truthful statements was marginally significant. No significant associations were observed for fabricated statements. This pattern is consistent with the original findings by Bogaard and Meijer [14], who also reported a small but significant correlation between accurate beliefs and credibility judgments of truthful statements ($r = .19$, $p < .004$), but no relationship for fabricated statements. However, the effect sizes in the current study were somewhat smaller. This may be due to the more skewed belief patterns observed: participants recognized fewer diagnostic cues correctly than in the original study, which may have limited the potential benefits of cue knowledge for veracity judgments. Still, the overall pattern aligns with prior work showing that knowledge of valid cues may selectively improve truth detection but not lie detection [15–17].

The repeated finding that beliefs correlate only with credibility scores of truthful statements may be explained by several factors. First, most empirically supported verbal cues signal truthfulness rather than deception [5,6], meaning accurate cue knowledge provides clearer diagnostic information for truth judgments. Second, individuals with relatively accurate cue knowledge may more readily recognize when statements conform to an expected truthful profile, thereby improving truth detection. In contrast, deceptive statements are more heterogeneous and less consistently marked by the absence of specific cues [3], making them inherently more difficult to identify even with accurate cue knowledge. Third, participants in our study showed poor accuracy in identifying diagnostic cues and frequently endorsed nondiagnostic features as indicative of deception. Consequently, when evaluating lies, they may have relied on invalid or misleading cues, which would limit, or even undermine, any potential benefit of their partial cue knowledge for lie detection.

In testing Hypothesis 2, the results provided no evidence that presentation influenced truth/lie discrimination accuracy or credibility judgments. Although previous research has reported that audio-visual formats reduce accuracy compared to audio-only or transcripts [9,11], we found no such effect in the current study. Furthermore, in contrast to Evanoff, Porter [13], who reported truth biases in transcripts, our data did not reveal any modality-based differences in either direction. This inconsistency across studies may reflect methodological differences, such as the type of stimuli used (high-stake emotional lies vs. negative autobiographical statements). Interestingly, Bogaard and Meijer [14] reported a truth bias in transcripts only among police officers but not lay participants, aligning with the present findings. Overall, our findings suggest that the modality in which statements are presented does not systematically influence accuracy or credibility, at least in the context of short autobiographical narratives.

A more robust finding across our data was that participants were significantly more accurate in detecting truths than lies. This is a well-established pattern in the literature and reflects the so-called "truth bias" [4], whereby people are more likely to judge statements as truthful, regardless of their veracity. Importantly, this effect held across all modalities, suggesting that the modality did not moderate participants' inherent bias toward believing others.

These findings have important implications for deception detection training and practice. Our results showed that cue knowledge showed only modest improvements in truth – but not lie – detection. Consequently, training programs focused simply on increasing cue knowledge will likely bring only limited improvement to lie detection accuracy. Further improvements may come from approaches that go beyond assessing behavioral or verbal cues in isolation. For example, previous research has shown that information-gathering protocols [24] and strategic questioning techniques [25,26] elicit additional details or create opportunities for checking statements against independent evidence.

Some limitations should be noted. First, although our sample was diverse in terms of modality assignment, the subset of participants who held a high number of accurate cue beliefs may have been too small to detect a strong effect. Second, the deception stimuli used consisted of brief, structured autobiographical statements. While this design ensured control and comparability, it may not fully capture the complexities of deception in real world deception detection contexts. Furthermore, the short duration may have reduced the potential for visual cues to act as (non-diagnostic) distractors, potentially explaining the lack of modality effects. As such, caution is warranted when generalizing these findings to applied settings. Third, participants completed the cue belief measure before the lie detection task, which may have primed their

attention to specific cues. However, this procedure was consistent with the original study. Finally, our measure of cue knowledge was based on self-report endorsement of predefined cues.

In conclusion, this study partially replicated Bogaard and Meijer's [1] findings. We found weak but suggestive evidence that accurate beliefs about verbal cues may aid in detecting truthful statements, though not deceptive statements. Contrary to our expectations, modality did not influence accuracy or credibility judgments, calling into question prior assumptions that visual input impairs deception detection. Further work is needed to clarify under which conditions accurate cue knowledge and modality translate into better performance, and whether that effect can be strengthened through training or feedback.

## Supporting information

**S1 Appendix. List of verbal criteria and their description.**
(DOCX)

## Author contributions

**Conceptualization:** Glynis Bogaard, Ewout H. Meijer.

**Data curation:** Sarah Elizabeth Fieweger.

**Formal analysis:** Glynis Bogaard, Sarah Elizabeth Fieweger.

**Investigation:** Sarah Elizabeth Fieweger.

**Methodology:** Glynis Bogaard, Ewout H. Meijer.

**Supervision:** Glynis Bogaard, Ewout H. Meijer.

**Writing – original draft:** Glynis Bogaard.

**Writing – review & editing:** Sarah Elizabeth Fieweger, Ewout H. Meijer.

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
