## [Decision Letter · Decision Letter 0]

23 Oct 2025

Dear Dr. Bogaard,

Thank you for submitting your manuscript to PLOS ONE. After careful consideration, we feel that it has merit but does not fully meet PLOS ONE’s publication criteria as it currently stands. Therefore, we invite you to submit a revised version of the manuscript that addresses the points raised during the review process.

We look forward to receiving your revised manuscript.

Kind regards,

Bartosz Wojciech Wojciechowski, Ph.D.

Academic Editor

PLOS ONE

2. In the online submission form, you indicated that [The data underlying this study are deposited in Dataverse: https://doi.org/10.34894/CY17PX. In accordance with institutional policy, these data are not openly accessible but can be obtained upon reasonable request.].

Additional Editor Comments:

Dear Glynis

Thank you for submitting your manuscript examining the relationship between beliefs about verbal cues and truth/lie discrimination accuracy. After careful consideration of the reviewers’ comments, I am pleased to inform you that your manuscript is considered suitable for publication, pending minor revisions.

Both reviewers commend the clarity, structure, and methodological rigor of your work. The replication of Bogaard and Meijer’s findings is executed with care and precision, and your transparent reporting—including preregistration and justification of minor deviations—adds to the credibility of the study. The reviewers particularly appreciated your operationalization of variables, sample size estimation, and pretesting procedures.

However, several points require your attention before the manuscript can be accepted:

Major Points for Revision

Clarify Experimental Design Early in the Method Section

Please specify the 2×3 design (Modality × Veracity) clearly at the beginning of the Method section to improve reader comprehension.

Stimuli Duration and Applied Value

Reviewer 1 raised concerns about the brevity of the video stimuli and its implications for applied contexts. While this does not undermine the replication value, consider briefly addressing this limitation in the Discussion section, particularly in relation to ecological validity.

Cue Belief Reporting and Table Presentation

Reviewer 2 found the reporting of cue belief results difficult to follow. Please:

Clarify the diagnostic vs. non-diagnostic cue recognition (e.g., “Of the 10 diagnostic cues, participants correctly identified X…”).

Revise the table to:

Use * to indicate statistical significance.

Group cues by their actual diagnostic category (e.g., “more for liars,” “inconclusive,” “less for liars”).

Add a column showing participants’ perceived cue directionality.

Verbal Cue Details

Include a more detailed description of the 17 verbal cues used in the study: their source, classification, and rationale for diagnosticity. This will help readers unfamiliar with the original study.

Believability Ratings

Provide more information on how the 1–5 believability scale was used in norming: What were the mean ratings? What thresholds were applied?

Discussion of Asymmetry in Cue Effects

Reviewer 2 suggests elaborating on why cue beliefs affected truth recognition but not lie detection. Consider discussing whether the nature or directionality of cues might explain this asymmetry and what implications this has for deception detection theory.

Minor Points

Clarify the reference to the “original study” in the abstract by naming Bogaard and Meijer explicitly.

Include examples of verbal deception cues earlier in the manuscript (e.g., p.4, line 90).

Address the possibility that cue rating tasks may have influenced participants’ judgments.

Consider including a comparative table showing how cue belief responses align or differ from Bogaard and Meijer’s findings.

Terminology Consistency

Reviewer 1 noted inconsistencies in terminology (e.g., “Audio” vs. “Transcript”). Please ensure consistent use of terms throughout the manuscript, especially in the Results section.

With best regards,

Bartosz

Reviewers' comments:

Reviewer's Responses to Questions

**Comments to the Author**

1. Is the manuscript technically sound, and do the data support the conclusions?

Reviewer #1: Yes

Reviewer #2: Yes

2. Has the statistical analysis been performed appropriately and rigorously?

Reviewer #1: Yes

Reviewer #2: Yes

3. Have the authors made all data underlying the findings in their manuscript fully available?

Reviewer #1: Yes

Reviewer #2: No

4. Is the manuscript presented in an intelligible fashion and written in standard English?

Reviewer #1: Yes

Reviewer #2: Yes

Reviewer #1: The study addresses the vital question of whether knowledge about valid verbal cues can improve truth/lie discrimination by attempting to replicate and extend previous findings on this relationship

The manuscript is well written, clear, and engaging. The authors present their ideas in a precise and well-structured manner. The argumentation is coherent and it is concise and to the point — it conveys all the necessary information without being overly wordy or redundant. I also greatly appreciate the strong methodological rigor demonstrated in this manuscript. The authors should be commended for their careful approach to study design, including the use of formal sample size estimation, very precise operationalization of the variables, pretesting video recordings and good and clear presentation of results. Of course ideally all of this should be considered a baseline standard for research in the detection deception area. Unfortunately, I have seen many studies with considerably weaker methodological practices, so this level of rigor is both commendable and refreshing.

I was a little surprised by the lack of modality effect on truth/lie discrimination accuracy or credibility judgments. I believe that it might be - as Authors wrote, due to stimuli. And stimuli is my main and only concern of this study. The stimuli utilized in this study were notably brief. Specifically, the mean duration of the video statements was 95.39 seconds, which is roughly 1.5 minutes. This restricted duration, coupled with the structured nature of the autobiographical statements, suggests that the viewing time might have been insufficient for many non-diagnostic visual distractors - such as nervous behaviors often stereotypically associated with lying - to fully emerge or influence judgment consistently. As a result, this brevity may have prevented a reduction in deception detection accuracy in the visual modality compared to the transcript-based condition. This suggests that the study has rather limited applied value (for example, for forensic or investigative contexts), although it remains valuable as a careful replication of the established relationship between beliefs about deception cues and detection accuracy.

While the preregistration and the final manuscript are highly consistent, there were very small and justified differences that reflect a refinement in the reporting of the research. I appreciate the authors’ explanation of those differences as this point initially caught my attention and was my point of concern. Their clarification demonstrates transparency and strengthens the credibility of the research process.

I strongly advise reviewing the final text once more, specifically focusing on consistency in terminology and minor reporting details. For example Authors In the Results section, when describing the ANOVA analysis state that a 3 (Modality: Audio, Visual, Audiovisual) × 2 (Veracity: Truth vs. Lie) analysis was conducted. however in all manuscript they described Audio, Transcript, and Audiovisual design. There are also several other minor errors of this kind, which, although small, make the text more difficult to read and follow.

Overall, the manuscript is very well written and methodologically sound. It represents a solid and careful piece of work, and the replication is executed with commendable rigor. That being said, in my opinion, while the study is of high quality, it does not contribute substantially new insights to the psychology of deception. In my view, the paper is a valuable replication study with a little “twist” that strengthens the empirical foundation of the field, but it is unlikely to change current theoretical perspectives or significantly advance the discipline.

Overall, I recommend the manuscript for publication, provided that the journal is open to publishing strong replications, even if they do not offer major conceptual innovation.

Reviewer #2: The authors conducted a single study extension of prior work based off Bogaard and Meijer’s previous study. Consistent to past work, they found that verbal cue beliefs had a small positive correlation for recognizing truthful statements. There was no effect on deceptive statements. There was no effect of statement modality on participants’ judgments.

The findings of this study are not particularly surprising. They replicate Bogaard and Meijer’s finding with their results on their binary truth/lie judgment and find a similar lack of effect due to statement modality seen frequently in the literature. I defer to the action editor to decide if that is novel enough to warrant publication. Other than that, I found the study to be soundly conducted and the results and conclusions reasonable. There are several things that would strengthen this article:

• Please state your design early in the method. The authors mention the 2x3 but please specify exactly what that is. They clarify in the results, but would be helpful to have that information upfront

• Please provide more information on how the norming ratings on the 1-5 believability scale were used. What were the means? What values were you looking for? Etc.

• Please provide more information on the 17 verbal cues, state what they are, where you got them from, the classifications, who is deciding cues are diagnostic, etc. I went and checked out the original paper, but readers should not have to chase down prior papers to evaluate and understand the study

• I found the reporting of cue belief results to be a bit harder to follow than expected (p10 starting on line 226). It would be helpful in the results section if the authors could pivot to something like “of the 10 diagnostic cues, they recognized X. Of the 7 nondiagnostic, they false alarmed to X.”

o For the table, I would swap the * to be indicative of significance as that is more common and then have the bold for it being accurate

o I would also group them in terms of the “Actual” column such that the “more for liars” are grouped together, then “inconclusive”, then “less for liars” or something to that effect

o Consider adding a column next to the “Actual” one that has the “Perceived” responses to make this easier to digest for your readers

• Why do the authors think there was only a small effect for true but not deceptive statements? Does the type of cues they were acc/inacc on provide insight? Or the directionality of the cues they were sensitive to? What does that mean for the field/implications, etc. Some elaboration about the implications of these data would strengthen the discussion.

Minor points:

• The abstract refers to an “original study”. It is clear what that is in the manuscript, but not clear prior to the introduction. I would just name the study in the abstract to clarify this

• It would be nice to include some examples of the verbal cues to deception on p4 line 90

• Is there any chance that having the participants provide their cue ratings first could have influenced their performance on the task? Maybe they considered something that they would not have if they had not read about it earlier?

• Did the cue belief responses replicate Bogaard and Meijer’s exactly? Any differences? A table for that could be interesting

**Do you want your identity to be public for this peer review?** For information about this choice, including consent withdrawal, please see our Privacy Policy

Reviewer #1: No

Reviewer #2: No

---

## [Author Response · Author response to Decision Letter 1]

2 Dec 2025

Please see our "response to reviewers" document

---

## [Editor Report · Decision Letter 1]

9 Dec 2025

I Told You So! Verbal Cue Beliefs Are Associated with Truth Detection, but Not Lie Detection

PONE-D-25-50292R1

Dear Dr. Bogaard,

We’re pleased to inform you that your manuscript has been judged scientifically suitable for publication and will be formally accepted for publication once it meets all outstanding technical requirements.

Kind regards,

Bartosz Wojciech Wojciechowski, Ph.D.

Academic Editor

PLOS ONE
---

## [Editor Report · Acceptance letter]

PONE-D-25-50292R1

PLOS One

Dear Dr. Bogaard,

I'm pleased to inform you that your manuscript has been deemed suitable for publication in PLOS One. Congratulations! Your manuscript is now being handed over to our production team.

Kind regards,

on behalf of

Dr. Bartosz Wojciech Wojciechowski

Academic Editor

PLOS One